# Using mHealth Technology to Evaluate Daily Symptom Burden among Adult Survivors of Childhood Cancer: A Feasibility Study

**DOI:** 10.3390/cancers16172984

**Published:** 2024-08-27

**Authors:** Kristen E. Howell, Jessica L. Baedke, Farideh Bagherzadeh, Aaron McDonald, Paul C. Nathan, Kirsten K. Ness, Melissa M. Hudson, Gregory T. Armstrong, Yutaka Yasui, I-Chan Huang

**Affiliations:** 1Department of Epidemiology and Biostatistics, School of Public Health, Texas A&M University, College Station, TX 77843, USA; khowell1@exchange.tamu.edu; 2Department of Epidemiology and Cancer Control, St. Jude Children’s Research Hospital, Memphis, TN 38105, USA; aaron.mcdonald@stjude.org (A.M.); kiri.ness@stjude.org (K.K.N.); greg.armstrong@stjude.org (G.T.A.); yutaka.yasui@stjude.org (Y.Y.); 3School of Public Health, University of Alberta, Edmonton, AB T6G 2R3, Canada; jessica.baedke@stjude.org (J.L.B.); bagherza@ualberta.ca (F.B.); 4Division of Haematology/Oncology, Hospital for Sick Children, Toronto, ON M5G 1X8, Canada; paul.nathan@sickkids.ca; 5Department of Oncology, St. Jude Children’s Research Hospital, Memphis, TN 38105, USA; melissa.hudson@stjude.org

**Keywords:** childhood cancer survivors, health-related quality of life, mHealth, momentary assessment, symptoms

## Abstract

**Simple Summary:**

Survivors of childhood cancer are predisposed to a range of late effects of treatment, including high symptom burden. Symptom burden may vary over time. The aim of this prospective pilot study was to assess the feasibility of collecting daily symptoms from adult survivors of childhood cancer using mobile health (mHealth) technology to evaluate symptom fluctuation and associations with future health-related quality of life (HRQOL). The 41 survivors included in this study had high adherence to symptom reporting (83%) and HRQOL reporting (95%). Variability of daily symptom burden differed from person-to-person (74%), day-to-day (18%), and month-to-month (8%). Additionally, a higher symptom burden was associated with poorer HRQOL in the future. Daily assessment of symptoms using mHealth technology revealed fluctuations in symptomology and the association between symptom burden and HRQOL. This method of symptom assessment is valuable for improving our understanding of symptom dynamics and sources of variability.

**Abstract:**

Background: Cancer therapies predispose survivors to a high symptom burden. This study utilized mobile health (mHealth) technology to assess the feasibility of collecting daily symptoms from adult survivors of childhood cancer to evaluate symptom fluctuation and associations with future health-related quality-of-life (HRQOL). Methods: This prospective study used an mHealth platform to distribute a 20-item cancer-related symptom survey (5 consecutive days each month) and an HRQOL survey (the day after the symptom survey) over 3 consecutive months to participants from the Childhood Cancer Survivor Study. These surveys comprised a PROMIS-29 Profile and Neuro-QOL assessed HRQOL. Daily symptom burden was calculated by summing the severity (mild, moderate, or severe) of 20 symptoms. Univariate linear mixed-effects models were used to analyze total, person-to-person, day-to-day, and month-to-month variability for the burden of 20 individual symptoms. Multivariable linear regression was used to analyze the association between daily symptom burden in the first month and HRQOL in the third month, adjusted for covariates. Results: Out of the 60 survivors invited, 41 participated in this study (68% enrollment rate); 83% reported their symptoms ≥3 times and 95% reported HRQOL in each study week across 3 months. Variability of daily symptom burden differed from person-to-person (74%), day-to-day (18%), and month-to-month (8%). Higher first-month symptom burden was associated with poorer HRQOL related to anxiety (regression coefficient: 6.56; 95% CI: 4.10–9.02), depression (6.32; 95% CI: 3.18–9.47), fatigue (7.93; 95% CI: 5.11–10.80), sleep (6.07; 95% CI: 3.43–8.70), pain (5.16; 95% CI: 2.11–8.22), and cognitive function (–6.89; 95% CI: –10.00 to –3.79) in the third month. Conclusions: Daily assessment revealed fluctuations in symptomology, and higher symptom burden was associated with poorer HRQOL in the future. Utilizing mHealth technology for daily symptom assessment improves our understanding of symptom dynamics and sources of variability.

## 1. Introduction

In the modern era, >85% of children diagnosed with cancer will become five-year survivors. However, the very therapies that cure pediatric cancers also predispose survivors to a range of late effects, such as physical and psychological sequelae, chronic health conditions, subsequent neoplasms, and premature death [1,2,3,4]. Additionally, a significant proportion of survivors experience poor patient-reported outcomes (PROs), including various symptoms [5] (e.g., pain, abnormal sensations, memory problems, somatization, pulmonary symptoms, and cardiac symptoms) and impaired health-related quality of life (HRQOL) [6]. Over 75% of childhood cancer survivors have reported experiencing multiple symptoms [5], and 50% of survivors experience moderate or high severity symptom burden [7]. A more severe symptom burden among childhood cancer survivors has been significantly associated with clinically ascertained physical and cognitive performance deficits and poorer HRQOL [7]. Importantly, previous research has reported the association between experiencing a greater symptom burden and an increased risk of mortality [8]. 

Symptom reporting is clinically relevant for individuals with cancer because the assessment can identify early signs of adverse events and improve patient–clinician communication for decision-making [9,10,11,12]. Conventional symptom assessments require participants to recall their symptom experiences over an extended period [i.e., seven days using the Patient Reported Outcomes Measurement Information System (PROMIS) or four weeks using the Short-Form 36 Health Survey Questionnaire (SF-36)], which yields averaged symptom information and introduces a recall bias. Moreover, symptom burden may vary over time, especially among individuals with a high disease burden. However, there is limited research exploring both within- and between-person variability in symptom experiences among cancer survivors. Assessing the rhythm of symptom change by considering within-person variability may serve as an indication of a change in health condition and suggest further clinical assessment and interventions. 

The application of ecological momentary assessment (EMA) allows for the real-time reporting of symptoms, HRQOL, and other PROs [13]. EMA is less subject to recall bias than retrospective surveys and recognizes fluctuations in symptom severity [14,15,16]. With the increasing accessibility of mobile health (mHealth) technology, EMA can be rapidly conducted through handheld devices such as tablets and smartphones [17,18]. For cancer survivors, leveraging mHealth technology to monitor symptoms and other PROs in real-time provides promising opportunities to detect adverse health events between clinic visits (or before the annual appointment) and to offer timely interventions [13,19,20]. Previous studies have identified barriers to using mHealth for PRO assessments or clinical interventions, such as protecting personal health information and technology challenges (i.e., unreliable Wi-Fi connection or application malfunction) [21,22,23]. Barriers to mHealth implementation may lead to poor compliance with data collection and healthcare engagement [21,24,25]. 

The present study evaluated the feasibility and adherence status of using an mHealth platform to collect daily symptom data and assessed the association between the fluctuation of symptom burden and future HRQOL over a 3-month window among adult survivors of childhood cancer. To improve adherence to daily symptom reporting, we applied technology-driven methods to communicate with participants throughout the study. Additionally, we estimated the within- and between-person variability for the severity of 20 individual symptoms relevant to the late effects of cancer therapies and tested the association of daily symptom burden with future HRQOL. Furthermore, we evaluated participants’ satisfaction with using the mHealth platform for daily symptom assessment and the potential of using mHealth-based daily symptom reports to communicate with healthcare providers for late-effect management. 

## 2. Methods

### 2.1. Study Design

This prospective study used an internet-based, mobile-enabled platform to examine 20 symptoms relevant to the late effects of cancer therapies and HRQOL among adult survivors of childhood cancer who were enrollees of the Childhood Cancer Survivor Study (CCSS). Participants self-reported their symptoms for 5 consecutive days per week on weeks 1, 5, and 9 over 3 consecutive months (Figure 1). At the end of each week, participants completed an HRQOL assessment. To prompt the PRO report, automatic text and e-mail reminders were sent for the daily symptom survey (5 days) and weekly HRQOL assessment (once per week) during weeks 1, 5, and 9, using a participant interface platform. The automatic communications included links to the online surveys that participants were required to complete that day.

### 2.2. Data Collection

Eligible survivors included those who had received a diagnosis of cancer before 18 years of age, exhibited survival >5 years after cancer diagnosis with no history of relapse, and attained an age of ≥18 years at the time of enrollment. Data collection occurred from May 2019 to October 2019. A random sample of 60 survivors meeting the criteria were invited to participate, with 20 from each of the three symptom burden groups (low, moderate, and high). Per the methods of our previous publication [7], data from 10 symptom domains, derived from 37 symptom items present in CCSS, were used to categorize eligible participants into low, moderate, and high symptom burden associated with physical and mental summary of HRQOL. Specifically, we categorized survivors as low burden if they reported symptoms in 0–1 domains [indicating normal HRQOL; 0.5 standard deviation (SD) above the norm], moderate burden if they reported symptoms in 2–5 domains (indicating moderate HRQOL impairment; 0.5–0.99 SD below the norm), and high burden if they reported symptoms in 6 or more domains (indicating high HRQOL impairment; 1 SD below the norm). 

### 2.3. PRO Assessment

This study evaluated 20 symptoms commonly experienced by childhood cancer survivors. These symptoms were selected based on data from previous research [5,7] and an opinion survey of 15 clinicians from St. Jude Comprehensive Cancer Center, who provide clinical and survivorship care for childhood cancer patients and survivors. In the symptom survey, if study participants reported a symptom as present in the past 24 h, they were asked to indicate the severity (mild, moderate, or severe) of that symptom during the same time frame. The 20 individual symptoms were irritability, anxiety, depression, fatigue or feeling weak, difficulty falling or staying asleep at night, sleepiness during the day, poor memory, lack of concentration, shortness of breath, chest pain during physical exercise, numbness or tingling, problems with balance, headache, bodily pain, swelling, cramps, constipation, diarrhea, lack of appetite, and poor coordination.

The outcome variables of interest were participation rate, adherence rate, variability of symptom burden over time (15 days over a three-month period), and the longitudinal association of symptom burden in the first month with HRQOL in the third month. The *participation rate* was defined as the number of participants who enrolled in this study compared to those who were invited to participate. The *weekly adherence rate* of a given week was defined as completing at least three daily symptom surveys and an HRQOL report in the same week. *Overall adherence rate* was defined as completing at least 12 of the 18 surveys (≥3 symptoms surveys and an HRQOL assessment in each of the 3 months) distributed throughout the entire study. The weekly HRQOL assessment induced six domains from the PROMIS-29 Profile [26] and the Neuro-QOL Cognitive Function–Short Form [27]. We used the PROMIS-29 to assess PROs in the anxiety, depression, fatigue, sleep disturbance, and pain interference domains, with higher scores indicating poorer HRQOL. The Neuro-QoL Cognitive Function assessment contains a single domain of perceived cognitive functioning, with higher scores indicating higher function status. The domain scores were calculated using a T-metric with a mean of 50 and a standard deviation of 10.

### 2.4. Satisfaction Survey

At the end of the 3-month study, participants completed an eight-item satisfaction survey designed by this study to provide feedback on their study participation experience. Participants reported the degree (strongly agree, agree, neither agree nor disagree, disagree, strongly disagree) to which they agreed with the following statements: (1) “It was easy for me to complete brief daily symptom evaluations (e.g., a few days in a week) over the past 3 months”; (2) “I would be willing to take part in symptom evaluations on a regular basis to help doctors understand more”; (3) “I would be willing to take part in symptom evaluations 2–3 times per day to help doctors learn the symptom changes on a daily basis”; (4) “I am interested in taking part in a clinical trial to help doctors use my symptom data for advancing treatment strategies”; (5) “In the future studies, I would be interested in receiving a report after my symptom evaluations are done”; (6) “I am interested in discussing problematic symptoms with my oncologists or primary care physicians”; (7) “I am interested in learning skills for self-managing my problematic symptoms”; and (8) “I believe that effective symptom controls may improve my quality of life”.

### 2.5. Covariates

Participants’ basic sociodemographic, treatment, and chronic health conditions (CHCs) data were obtained from the CCSS database. Sociodemographic variables included attained age at time of the first survey, time since cancer diagnosis, sex, race/ethnicity, and educational attainment. Clinical variables included chemotherapy with anthracyclines, methotrexate, plant alkaloids, cyclophosphamide, bleomycin, cytarabine, and steroids; radiation to the brain, chest, or pelvis; and invasive surgery (amputation and limb-sparing surgery, nephrectomy, splenectomy, and exenteration and organ-preservation surgery). CHCs were assessed by seven organ systems representing 18 individual CHCs: (1) hearing (e.g., loss of hearing); (2) vision (e.g., cataracts, blindness); (3) endocrine (e.g., hypothyroid, thyroid nodules, diabetes, osteoporosis); (4) respiratory (e.g., emphysema); (5) cardiovascular (e.g., arrhythmia, hypertension, cholesterol); (6) renal (e.g., kidney stones, dialysis); and (7) neurological (e.g., memory problems, balance, tremors, weakness in leg, sensory neuropathy). The severity of individual CHCs was graded by the Common Terminology Criteria for Adverse Events, as used in previous CCSS publications [1,28].

### 2.6. Statistical Analysis

Statistical analysis was limited to those meeting the overall adherence criteria, requiring completion of at least 12 of the 18 surveys (i.e., ≥3 daily symptoms reports and 1 HRQOL report in each of the 3 weeks). For each participant, the burden of up to 15 daily symptom reports (5 days × 3 months) was calculated using the available symptom surveys. The daily burden across 20 symptoms was defined as the sum of the severity scores (none = 0, mild = 1, moderate = 2, severe = 3) for each of the 20 symptoms reported in the same week, yielding a range from 0 to 60. The monthly burden across 20 individual symptoms was defined as the average daily burden of these symptoms over 5 days for each of the three months, yielding a range from 0 to 60. Both daily and monthly burdens were standardized with division by the standard deviation of all monthly burdens, which was 7.20.

A total variation, consisting of between-person and within-person variations, was estimated for each of the 20 symptoms using the standardized values. Within-person variability was further decomposed into month-to-month and day-to-day variation. For each symptom variable, a univariate linear mixed model was conducted by incorporating symptom surveys collected over all three months. Nested random intercepts at both the person and month levels estimated person-to-person variance and month-to-month variance components. The residual variance was interpreted as day-to-day variation. These three variances, expressed as a fraction of the total variance, provided the variability in symptom burden attributable to the person-to-person, month-to-month, and day-to-day components. Symptom variability was analyzed for all participants and subgroups based on different cancer diagnoses. Diagnoses were categorized into hematological (acute lymphoblastic leukemia, non-Hodgkin lymphoma, and Hodgkin lymphoma), central nervous system (CNS) tumor, and solid tumor (Wilms tumor, neuroblastoma, rhabdomyosarcoma, osteosarcoma, and Ewing sarcoma).

The longitudinal association of symptom burden with future HRQOL was assessed with multivariable linear regression. The exposure variable (i.e., standardized monthly symptom burden across 20 individual symptoms) was calculated from the first month’s symptom survey, and the dependent variable (i.e., each of the six HRQOL domains) was calculated from the third month’s HRQOL survey. A separate model was performed for each of the HRQOL outcomes. Covariates adjusted in the analysis included attained age at first symptom survey, sex, and exposure to radiation, chemotherapy, and invasive surgery. 

Analysis was conducted in Stata version 14.2, R version 4.3.0, and Rstudio version 2023.03.2+454 [29,30,31].

## 3. Results

Out of 60 survivors invited to participate, 41 (68%) enrolled in the study (Table 1). Of the 41 survivors, 34 (83%) reported their symptoms at least three times each week over 3 months, and 39 (95%) reported their HRQOL each week over 3 months (Appendix A). Of the enrolled survivors, 66% were female; 73% were white, 59% were aged 30–39.9 years at the time of enrollment; and 68% had been diagnosed 20–29 years prior to the study (Table 1). Cancer diagnoses included CNS (37%), solid tumor (26%), leukemia (20%), and lymphoma (17%). Participants underwent cancer therapies including chemotherapy (61%), radiation (27%), and invasive surgery (78%). Nearly all participants (93%) had reported at least one chronic health condition prior to the study. 

Symptoms reported as having the highest prevalence in each month were difficulty falling/staying asleep, bodily pain, headache, and anxiety (Appendix A). The total burden across 20 symptoms over 3 months (i.e., the mean of symptom burdens over 3 months) among participants was 6.93 (Appendix A). Mean symptoms reported were higher among the survivors with arrhythmia (6.5 symptoms, 95% CI: 3.1–9.9), memory problem (2.7 symptoms, 95% CI: 0.8–4.5), and sensory neuropathy (5.9 symptoms, 95% CI: 2.7–9.1), each compared to those without it. The variance of total burden across 20 symptoms over 3 months was attributed person-to-person (74%), day-to-day (18%), and month-to-month (8%) (Figure 2). Six symptoms (poor memory, bodily pain, lack of concentration, irritability, anxiety, and difficulty sleeping) were more likely to vary from person-to-person (≥50% of the total variance). The other 14 symptoms (70%) exhibited within-person variability that accounted for >50% of the total variability. Cramps, diarrhea, poor coordination, shortness of breath, constipation, and chest pain during exercise were more likely to vary day-to-day (≥50% of the total variance). Bodily pain, irritability, anxiety, difficulty sleeping, headache, and fatigue had a high total variance of symptom burden (≥0.80). A supplemental analysis found that the diagnosis group did not greatly explain symptom variability (Appendix A). However, compared to the hematological and solid tumor survivors, CNS tumor survivors had smaller person-to-person symptom variability and larger month-to-month symptom variability.

A higher burden of 20 symptoms in the first month was significantly associated with poorer HRQOL in the third month, adjusting for age at week 1′s symptom survey, sex, and history of chemotherapy, radiation, and invasive surgery (Figure 3). Specifically, a one standard deviation increase in the burden of 20 symptoms in the first month was associated with poorer HRQOL scores for all domains in the third month, including 7.93 (95% CI: 5.11, 10.80) in fatigue interference, 6.56 (95% CI: 4.10, 9.02) in anxiety interference, 6.32 (95% CI: 3.18, 9.47) in depression interference, 6.07 (95% CI: 3.43, 8.70) in sleep interference, 5.16 (95% CI: 2.11, 8.22) in pain interference, and −6.89 (95% CI: −10.00, −3.79) in cognitive function.

Most participants (89.7%) reported that the daily activities were easy to complete and that they would be willing to participate in symptom evaluations on a daily basis (Table 2). Additionally, participants related that they would be interested in receiving a report of their symptom evaluations (82.1%), learning skills for self-managing problematic symptoms (87.2%), and discussing problematic symptoms with their providers (76.9%).

## 4. Discussion

This study found that the burden of individual symptoms among adult survivors of childhood cancer can vary from person-to-person and from day-to-day. The use of momentary symptom data collection typically facilitates capturing the frequent fluctuation in PROs and within-person and between-person variabilities of PROs, emphasizing the importance of conducting momentary symptom assessments to collect dynamic symptom data from this population who are at risk of high disease burden and premature mortality [3,4]. This study further demonstrates the feasibility of utilizing an mHealth platform to gather complex daily symptom data and reveals a significant association between daily symptom fluctuations and future HRQOL in adult survivors of childhood cancer. Participants also demonstrated high adherence to daily study activities, and most reported that these activities were easy to complete and expressed willingness to participate in a similar study in the future. Altogether, leveraging technology-based communication and data collection enhances the ability to capture momentary PRO data in large-scale research and clinical practice. 

This study was conducted before the onset of the COVID-19 pandemic, which has since significantly impacted global health. We expect that the symptom burden may have increased and quality of life may have declined as short-term effects of the pandemic due to heightened stress, healthcare disruptions, and social isolation. However, we believe that, in the long term, both symptom burden and quality of life will gradually return to pre-pandemic levels as conditions stabilize. Additionally, we anticipate an improved response rate in future studies, given the increasing adoption of mHealth devices and widespread availability of data plans for collecting patient-reported outcomes, which make remote participation more feasible and efficient.

The dynamic data collected via momentary assessments revealed that the burden of each symptom fluctuated both day-to-day and month-to-month within individual survivors, as well as between survivors. While 74% of the total variability in daily symptom burden was due to differences between individual survivors, the majority of the individual symptoms (14 out of 20) exhibited within-person variability that accounted for >50% of the total variability. This finding suggests the clinical importance of assessing specific symptoms more frequently (i.e., at least monthly), as these symptoms may change from day-to-day or from month-to-month. The conventional use of a one-time symptom assessment in clinical practice (i.e., one symptom assessment from an annual survivorship follow-up visit) would not fully capture the natural progression of these symptoms. 

An important goal of survivorship care for childhood cancer survivors, many of whom face multiple chronic health conditions, is to improve or maintain optimal HRQOL. The discovery of a significant association between higher daily symptom burden in an early month and poorer HRQOL in a later month underscores the importance of collecting momentary symptom data. This approach enables precise tracking of long-term symptom fluctuations, providing deeper insights into survivors’ well-being and functional status [32]. Providers could conduct a symptom burden assessment to monitor the progression of late effects and integrate this information into the design of clinical interventions targeting late-effect management. Future studies can utilize the mHealth approach for momentary data collection to examine the determinants of symptom burden and variability [7]. Recent findings suggest that regular symptom assessments can facilitate earlier diagnosis of cancer in adults [33]. Leveraging mHealth technology for this purpose provides promising opportunities to improve risk prediction of impaired health status and deliver timely, target interventions. 

Study participants reported high satisfaction with completing the daily symptom reporting and expressed a desire to learn more from their providers about managing problematic symptoms. The mHealth approach to collecting symptoms proved to be useful, enabling day-by-day data collection via technology, which is more efficient and reliable than the traditional, time-consuming paper-and-pencil-based method. However, it is important to acknowledge that the use of mHealth technology has its limitations due to issues of data security, inaccessibility of technology, and completeness of the data [34,35,36]. Though this study addressed data security concerns by using a secure server and messaging system, the reliance on technology (e.g., smartphones or tablets) may inherently be biased toward individuals with access to smartphones and network services (e.g., Wi-Fi or data plans). 

This feasibility study was limited to a small sample size. Therefore, we were unable to stratify the results by cancer diagnosis or treatment. Cancer diagnosis and/or treatment factors may affect compliance with study activities due to potential impacts on cognitive function (i.e., brain tumors or radiation) [37,38]. Additionally, this study did not consider seasonal variability or other environmental factors that may fluctuate the symptom burden throughout the year. Furthermore, this study did not consider psychosocial support that study participants may or may not have received, which may confound the association between symptom burden and HRQOL.

This study found that adult survivors of childhood cancer were adherent to completing frequent symptom assessments. Frequent symptom assessments further acknowledged the within-person and between-person variability in the burden of individual symptoms over a 3-month period. Future studies are warranted to implement frequent symptom monitoring and apply ecological momentary assessment through the mHealth platform to assess complex, dynamic symptom burdens for childhood cancer survivors. 

## Figures and Tables

**Figure 1 cancers-16-02984-f001:**
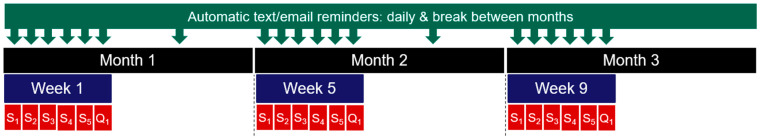
Study design and timeline. S = symptom assessment; Q = health-related quality of life assessment.

**Figure 2 cancers-16-02984-f002:**
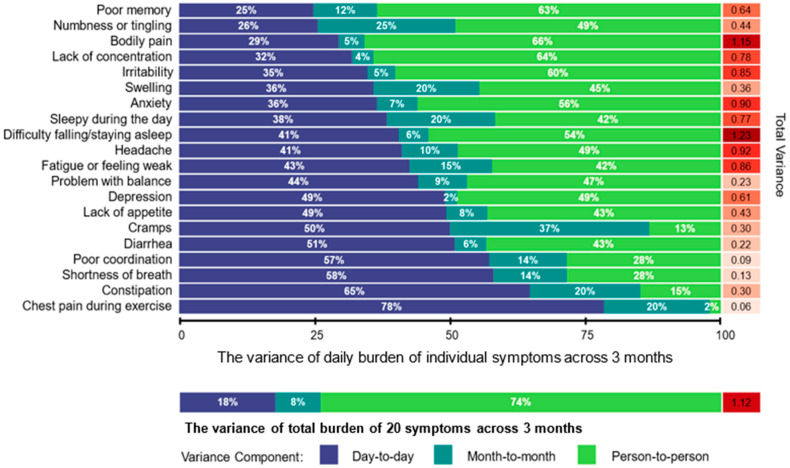
Variability of daily symptom burden across 3 months among 41 survivors.

**Figure 3 cancers-16-02984-f003:**
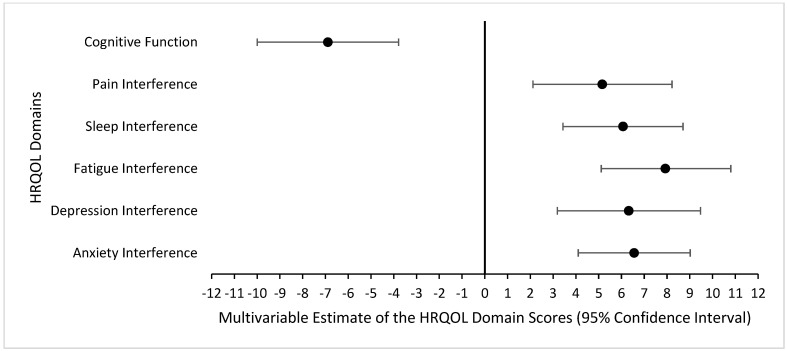
Multivariable regression coefficient for the burden of 20 symptoms in the first month associated with HRQOL in the third month. (1) The coefficients depict the degree to which the HRQOL T-score changed by one standard deviation with an increase in monthly symptom burden. (2) HRQOL was measured using the PROMIS-29 Profile and the Neuro-QOL. (3) Higher scores indicate worse HRQOL for all domains in the PROMIS-29 Profile, and better cognitive function in the Neuro-QOL. (4) Models are adjusted for attained age at symptom survey in week 1 of month 1, sex, and history of chemotherapy, radiation, and invasive surgery.

**Table 1 cancers-16-02984-t001:** Demographic and treatment characteristics of study participants (N = 41).

Characteristics	N (%) or Mean (SD; Min, Max)
Age at study enrollment (in years)	34.0 (5.3; 25.7, 47.1)
Age at study enrollment (%)	
18–29.9 years	10 (24)
30–39.9 years	24 (59)
≥40 years	7 (17)
Time since cancer diagnosis (in years)	26.1 (3.7; 19.5, 32.4)
Time since cancer diagnosis (%)	
10–19 years	3 (7)
20–29 years	28 (68)
≥30 years	10 (24)
Sex (%)	
Male	14 (34)
Female	27 (66)
Race/Ethnicity (%)	
White non-Hispanic	30 (73)
Black non-Hispanic	11 (27)
Educational Attainment	
High school/GED or less	6 (15)
Some college or post-high school training	21 (51)
College graduate or post-graduate level	14 (34)
Cancer diagnosis (%)	
Central nervous system tumor	15 (37)
Acute lymphoblastic leukemia	8 (20)
Wilms tumor	5 (12)
Non-Hodgkin lymphoma	5 (12)
Hodgkin lymphoma	2 (5)
Neuroblastoma	2 (5)
Rhabdomyosarcoma	2 (5)
Osteosarcoma	1 (2)
Ewing sarcoma	1 (2)
Cancer Treatment	
Chemotherapy (%)	25 (61)
Radiation (%)	11 (27)
Invasive surgery (%)	32 (78)
PROMIS-29 Profile reported in Month 3	
Anxiety Interference	56.3 (9.6; 40.3, 73.4)
Depression Interference	55.7 (10.4; 41.0, 79.3)
Fatigue Interference	54.5 (11.8; 33.7, 75.8)
Sleep Interference	56.0 (9.5; 36.9, 73.3)
Pain Interference	54.0 (10.9; 41.6, 75.6)
Neuro-QOL reported in Month 3	
Cognitive Function	47.1 (11.2; 22.8, 64.2)

**Table 2 cancers-16-02984-t002:** Satisfaction with daily symptom assessment and future use for survivorship care (N = 39).

Questions	Strongly Agree/Agree	Neutral	Strongly Disagree/Disagree
It was easy for me to complete brief daily symptom evaluations (e.g., a few days in a week) over the past 3 months	35 (89.7%)	2 (5.1%)	2 (5.1%)
I would be willing to take part in symptom evaluations on a regular basis to help doctors understand more	35 (89.7%)	3 (7.7%)	1 (2.6%)
I would be willing to take part in symptom evaluations 2–3 times per day to help doctors learn the symptom changes on a daily basis	26 (66.7%)	9 (23.1%)	4 (10.3%)
I am interested in taking part in a clinical trial to help doctors use my symptom data for advancing treatment strategies	31 (79.5%)	8 (20.5%)	0 (0%)
In future studies, I would be interested in receiving a report after my symptom evaluations are done	32 (82.1%)	5 (12.8%)	2 (5.1%)
I am interested in discussing problematic symptoms with my oncologists or primary care physicians	30 (76.9%)	9 (23.1%)	0 (0%)
I am interested in learning skills for self-managing my problematic symptoms	34 (87.2%)	5 (12.8%)	0 (0%)
I believe that effective symptom control may improve my quality of life	31 (79.5%)	7 (17.9%)	1 (2.6%)

## Data Availability

The Childhood Cancer Survivor Study is a U.S. National Cancer Institute-funded resource (U24CA55727) to promote and facilitate research among long-term survivors of cancer diagnosed during childhood and adolescence. CCSS data are publicly available on dbGaP at https://www.ncbi.nlm.nih.gov/gap/, accessed on 22 August 2024, through its accession number phs001327.v2.p1. and on the St. Jude Survivorship Portal within the St. Jude Cloud at https://survivorship.stjude.cloud/, accessed on 22 August 2024. In addition, utilization of the CCSS data that leverages the expertise of CCSS Statistical and Survivorship research and resources will be considered on a case-by-case basis. For this utilization, a research Application of Intent followed by an Analysis Concept Proposal must be submitted for evaluation by the CCSS Publications Committee. Users interested in utilizing this resource are encouraged to visit http://ccss.stjude.org, accessed on 22 August 2024. Full analytical data sets associated with CCSS publications since January of 2023 are also available on the St. Jude Survivorship Portal at https://viz.stjude.cloud/community/cancer-survivorship-community~4/publications, accessed on 22 August 2024.

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
