# Peer review of "Using mHealth Technology to Evaluate Daily Symptom Burden among Adult Survivors of Childhood Cancer: A Feasibility Study"

_cancers, 2024, doi:10.3390/cancers16172984_

Round 1

Reviewer 1 Report

Comments and Suggestions for Authors

Howell and colleagues present their data on the use of mobile health technology to evaluate daily symptom burden on 41 adult survivors of childhood cancer.  Overall the paper is well written and flows nicely.  Here are my comments:

1. The data was collected in 2019.  Is there any particular reason why this data is being presented now and do the authors expect the results to be any different in 2024, now 5 years later, especially as some of the symptoms queried relate to mental well being (sleep, depression, anxiety, etc) which has all increased in society following the global pandemic.  

2. Cancer diagnoses and treatment are shown in the first paragraph of the results.  What proportions of patients survived relapsed disease and did any receive allo-SCT? 

3. Minor comment. Table 1 should have the category headings left justified to make it easier to read. 

4. The person-to-person variability accounted for 74% of all symptom variability.  I imagine that some/most of this is due to the variety of diagnoses included.  For example, leukemia patients are very different than CNS patients who are different than MSK-tumor patients.  I recognize a limitation of the study is the small size and they were unable to stratify by diagnosis but what is the person-to-person variability within a single diagnostic category?  Without knowing this information, it's hard to interpret what a 74% person-to-person variability really means. 

5. Figure 3. I'm not sure what this Figure is showing and how to understand this.  "A one SD increase in burden of 20 symptoms in the first month is associated with poorer HRQOL scores for all domains in the 3rd month".   Is the increase in SD measured from Month 1 to Month 2, or from Day 1 to Day 5 during the first month?  How does one interpret the multivariable coefficient shown in Figure 3?  If these are odds ratios, please indicate that.  Please interpret the difference between a negative coefficient (cognitive function was -6.89) and that is different from the positive ones. 

Author Response

Overall Response:

Thank you for the opportunity to revise our manuscript.  In this response letter below, two reviewers’ comments are italicized, and our point-by-point response to each comment is written in regular font. In our revised manuscript, we used underlined, red-colored texts to indicate where changes have been made.

REVIEWER 1

Howell and colleagues present their data on the use of mobile health technology to evaluate daily symptom burden on 41 adult survivors of childhood cancer.  Overall the paper is well written and flows nicely.  Here are my comments:

  1. The data was collected in 2019. Is there any particular reason why this data is being presented now and do the authors expect the results to be any different in 2024, now 5 years later, especially as some of the symptoms queried relate to mental well being (sleep, depression, anxiety, etc) which has all increased in society following the global pandemic. 

 Response: The purpose of this pilot study was to determine the feasibility of the mHealth methods for collecting symptom reports on a day-to-day basis and to demonstrate the variability of symptoms from day-to-day. Data used in this manuscript were collected as part of a pilot study for a grant currently funded by the U.S. National Cancer Institute (NCI), and we believe it is valuable to publish the results. We anticipate that the symptom burden may be higher, and quality of life may be poorer following the global pandemic as a short-term effect. However, we anticipate that, in the long term, the symptom burden and quality of life will return to pre-pandemic levels. Nevertheless, we anticipate a higher response rate in the future given the growing popularity of mHealth devices and data plans for collecting patient-reported outcomes.     

In the revised manuscript, we addressed this point in the second paragraph of the Discussion section, which reads: “This study was conducted before the onset of the COVID-19 pandemic, which has since significantly impacted global health. We expect that the symptom burden may have increased, and quality of life may have declined as short-term effects of the pandemic due to heightened stress, healthcare disruptions, and social isolation. However, we believe that, in the long term, both symptom burden and quality of life will gradually return to pre-pandemic levels as conditions stabilize. Additionally, we anticipate an improved response rate in future studies, given the increasing adoption of mHealth devices and widespread availability of data plans for collecting patient-reported outcomes, which make remote participation more feasible and efficient.”

  1. Cancer diagnoses and treatment are shown in the first paragraph of the results. What proportions of patients survived relapsed disease and did any receive allo-SCT?

 Response: We excluded all patients with relapse. We have added this point to the Methods section under the header “Data Collection”.

  1. Minor comment. Table 1 should have the category headings left justified to make it easier to read.

 Response: Thank you. We will ask for this to be adjusted in the proof.

  1. The person-to-person variability accounted for 74% of all symptom variability. I imagine that some/most of this is due to the variety of diagnoses included.  For example, leukemia patients are very different than CNS patients who are different than MSK-tumor patients.  I recognize a limitation of the study is the small size and they were unable to stratify by diagnosis but what is the person-to-person variability within a single diagnostic category?  Without knowing this information, it's hard to interpret what a 74% person-to-person variability really means.

 Response: We conducted additional analyses for symptom variability by stratifying survivors into three diagnosis groups: hematological (acute lymphoblastic leukemia, non-Hodgkin lymphoma, and Hodgkin lymphoma), central nervous system (CNS) tumors, and solid tumor (Wilms tumor, neuroblastoma, rhabdomyosarcoma, osteosarcoma, and Ewing sarcoma). This additional analysis suggests that 1) the variation of symptoms was not greatly explained by diagnosis groups and 2) compared to the hematological and solid tumor survivors, CNS tumor survivors had a smaller person-to-person variation and larger month-to-month variation of symptoms (see data in the following table). 

Diagnosis Group

Variance Component

Variance

Variance (%)

Total Variance

Hematological

Person-to-person

0.96

77.8%

1.23

Month-to-month

0.06

5.2%

Day-to-day

0.21

16.9%

CNS

Person-to-person

0.65

65.8%

0.99

Month-to-month

0.14

14.4%

Day-to-day

0.12

19.8%

Solid tumor

Person-to-person

0.84

76.8%

1.09

Month-to-month

0.07

6.0%

Day-to-day

0.19

17.2%

All diagnoses

Person-to-person

0.83

74.0%

1.12

Month-to-month

0.09

8.3%

Day-to-day

0.20

17.7%

We have added this approach to the Methods section under the header “Statistical Analysis”, which now reads: “Symptom variability was analyzed for all participants and subgroups based on different cancer diagnoses. Diagnoses were categorized into hematological (acute lymphoblastic leukemia, non-Hodgkin lymphoma, and Hodgkin lymphoma), central nervous system (CNS) tumor, and solid tumor (Wilms tumor, neuroblastoma, rhabdomyosarcoma, osteosarcoma, and Ewing sarcoma).” Additionally, we reported these findings in the second paragraph of the Results section, which now reads: “A supplemental analysis found that the diagnosis group did not greatly explain symptom variability (Supplemental Table 2). However, compared to the hematological and solid tumor survivors, CNS tumor survivors had smaller person-to-person symptom variability and larger month-to-month symptom variability.” Finally, we summarized these new findings in a new table (see Supplemental Table 2).

  1. Figure 3. I'm not sure what this Figure is showing and how to understand this. "A one SD increase in burden of 20 symptoms in the first month is associated with poorer HRQOL scores for all domains in the 3rd month".   Is the increase in SD measured from Month 1 to Month 2, or from Day 1 to Day 5 during the first month?  How does one interpret the multivariable coefficient shown in Figure 3?  If these are odds ratios, please indicate that.  Please interpret the difference between a negative coefficient (cognitive function was -6.89) and that is different from the positive ones.

 Response: The purpose of Figure 3 is to demonstrate the modeled associations between symptom burden and future HRQOL. These are estimated temporal associations rather than causal relationships. We are not measuring a change in symptom burden, instead, we are assessing the overall symptom burden reported in Month 1 and its association with HRQOL reported in Month 3. The coefficients are based on the multivariable linear regression models, which estimate how higher symptom burden (for every 1 SD increase in burden) is associated with poorer HRQOL.

In this study, HRQOL was assessed using the PROMIS-29 (higher scores indicate poorer HRQOL) and the Neuro-QOL (higher scores indicate better function status). Cognitive function was on the Neuro-QOL and was therefore scored opposite compared to the domains on the PROMIS. This issue was described in the original manuscript in the Methods section under the header “PRO Assessment”.

Reviewer 2 Report

Comments and Suggestions for Authors

The study is interesting, innovative and methodologically well conducted.

This present study focuses on the feasibility and adherence status of using a mHealth platform to collect daily symptom data in order to define the association between the fluctuation of different daily symptoms and future HRQOL over a 3-month window among adult survivors of childhood cancer.

The feasibility of the use of this electronic tool is well described in the results and discussion however some points could be highlighted.

1. In the covariates section the authors should better detail ‘ invasive surgery’. As they detailed chemotherapy and radiotherapy, the site and type of surgery specification could be useful to better understand the patient cohort.

2. Since the results highlight a burden that is mainly emotional (e.g., anxiety, irritability) and physical (e.g., fatigue), it would be interesting to know if the participants were taken care of from a psychological and physical-motor point of view during the survivorhip care. These information could explain some results and they could help clinicians in the proposal of dedicated intervention during treatment and survivorship phase.

3. There are certainly some limitations clearly highlighted by the researchers that will have to be taken into consideration for the continuation of the study.

To have more information from this study it coul be useful to know if at the time of the study the patients present of 1 or more chronic health conditions 

Furthermore it could be interesting to know if a significant variation in symptoms in the timeframe corresponds to a significant variation in any chronic health condition

Author Response

Overall Response:

Thank you for the opportunity to revise our manuscript.  In this response letter below, two reviewers’ comments are italicized, and our point-by-point response to each comment is written in regular font. In our revised manuscript, we used underlined, red-colored texts to indicate where changes have been made.

REVIEWER 2

The study is interesting, innovative and methodologically well conducted. This present study focuses on the feasibility and adherence status of using a mHealth platform to collect daily symptom data in order to define the association between the fluctuation of different daily symptoms and future HRQOL over a 3-month window among adult survivors of childhood cancer.  The feasibility of the use of this electronic tool is well described in the results and discussion however some points could be highlighted.

  1. In the covariates section the authors should better detail ‘ invasive surgery’. As they detailed chemotherapy and radiotherapy, the site and type of surgery specification could be useful to better understand the patient cohort.

 Response: The invasive surgery includes amputation, amputation and limb-sparing surgery, nephrectomy, splenectomy, and exenteration and organ-preservation surgery. We have added these details to the Methods section under the header “Covariates.”

  1. Since the results highlight a burden that is mainly emotional (e.g., anxiety, irritability) and physical (e.g., fatigue), it would be interesting to know if the participants were taken care of from a psychological and physical-motor point of view during the survivorship care. This information could explain some results, and they could help clinicians in the proposal of dedicated intervention during the treatment and survivorship phase.

Response: This study did not collect information on the psychosocial support that patients may have received. We agree that psychosocial support may confound the association between symptom burden and HRQOL and have included this point in the Limitations paragraph of the Discussion section, which now reads: “Furthermore, this study did not consider psychosocial support that study participants may or may not have received, which may confound the association between symptom burden and HRQOL.”

  1. There are certainly some limitations clearly highlighted by the researchers that will have to be taken into consideration for the continuation of the study.

 Response: The sample size for this pilot study was small, but it demonstrated the feasibility of using this method for a larger-scale study currently funded by the U.S. National Cancer Institute.  

  1. To have more information from this study it could be useful to know if at the time of the study the patients present of 1 or more chronic health conditions.

Response: In this study, 38 out of 41 survivors had at least 1 chronic health condition, and 18 types of individual chronic health conditions were evaluated. This point has been included in the first paragraph of the Results section, which now reads: “Nearly all participants (93%) had reported at least one chronic health condition prior to the study.”

  1. Furthermore, it could be interesting to know if a significant variation in symptoms in the timeframe corresponds to a significant variation in any chronic health condition.

Response: We investigated the variation of symptoms by the presence of chronic health conditions at the same time and reported this finding in the second paragraph of the Results section, which now reads: “Mean symptoms reported were higher among the survivors with arrhythmia (6.5 symptoms, 95%CI 3.1-9.9), memory problem (2.7 symptoms, 95% CI 0.8-4.5), and sensory neuropathy (5.9 symptoms, 95% CI 2.7-9.1), each compared to those without it.”
